# MicroRNA-181c Inhibits Interleukin-6-mediated Beta Cell Apoptosis by Targeting TNF-α Expression

**DOI:** 10.3390/molecules24071410

**Published:** 2019-04-10

**Authors:** Yoon Sin Oh, Gong Deuk Bae, Eun-Young Park, Hee-Sook Jun

**Affiliations:** 1Department of Food and Nutrition, Eulji University, Seongnam 13135, Korea; 2Lee Gil Ya Cancer and Diabetes Institute, Gachon University, Incheon 21999, Korea; chocoba819@naver.com; 3Natural Medicine Research Institute, College of Pharmacy, Mokpo National University, Jeonnam 58554, Korea; parkey@mokpo.ac.kr; 4Gachon Institute of Pharmaceutical Science, College of Pharmacy, Gachon University, Incheon 21936, Korea

**Keywords:** Interleukin-6, beta cell apoptosis, microRNA-181c

## Abstract

We have previously reported that long-term treatment of beta cells with interleukin-6 (IL-6) is pro-apoptotic. However, little is known about the regulatory mechanisms that are involved. Therefore, we investigated pro-apoptotic changes in mRNA expression in beta cells in response to IL-6 treatment. We analyzed a microarray with RNA from INS-1 beta cells treated with IL-6, and found that TNF-α mRNA was significantly upregulated. Inhibition of TNF-α expression by neutralizing antibodies significantly decreased annexin V staining in cells compared with those treated with a control antibody. We identified three microRNAs that were differentially expressed in INS-1 cells incubated with IL-6. In particular, miR-181c was significantly downregulated in IL-6-treated cells compared with control cells and the decrease of miR-181c was attenuated by STAT-3 signaling inhibition. TNF-α mRNA was a direct target of miR-181c and upregulation of miR-181c by mimics, inhibited IL-6-induced increase in TNF-α mRNA expression. Consequently, reduction of TNF-α mRNA caused by miR-181c mimics enhanced cell viability in IL-6 treated INS-1 cells. These results demonstrated that miR-181c regulation of TNF-α expression plays a role in IL-6-induced beta cell apoptosis.

## 1. Introduction

Prolonged and uncontrolled activation of inflammation is a hallmark of many diseases, such as obesity-associated metabolic disorders including type 1 and type 2 diabetes [1]. Nutritionally overload cells triggers obesity induced inflammation, which is elevated expression of cytokines, chemokines and inflammatory mediators from metabolic organs and results in systemic inflammatory responses and disrupt glucose homeostasis [2].

Interleukin (IL)-6 is a pleiotropic cytokine that is associated with several immunoinflammatory and autoimmune diseases such as systemic lupus erythematous, rheumatoid arthritis and diabetes [3,4,5] and is produced by a variety of different cell types with complex, cell-specific mechanisms [6]. Expression of IL-6 is elevated during obesity and positively correlates with subsequent development of type 1 and 2 diabetes [7,8]. Moreover, antibody-mediated neutralization of IL-6 in genetically obese mice acutely improves hepatic insulin activity [9] and anti-IL-6 antibody treatment showed significantly reduced incidence of diabetes with decreased insulitis in autoimmune diabetic nonobese diabetic (NOD)/Walter & Eliza Hall Institute (Wehi) mouse [5].

Previously, we found that chronic treatment of beta cells (INS-1) with IL-6 decreased cell viability. Additionally we found that signal transducer and activator of transcription-3 (STAT-3) signaling was involved in IL-6-induced beta cell apoptosis [10]. To identify potential pro-apoptotic mechanisms during IL-6 treatment, we investigated changes in apoptotic pathway gene expression profiles using PCR microarrays.

Tumor necrosis factor alpha (TNF-α) is a proinflammatory cytokine with pleiotropic effects that are mediated through cell surface receptors, TNF receptor 1 (TNFR1) and TNF receptor 2 (TNFR2). TNFR1 has a death domain in its cytoplasmic region [11]. TNFR1-linked apoptotic factors, such as TNFR1-associated death domain protein (TRADD), Fas-associated protein with death domain (FADD), and FADD-like interleukin-1β-converting enzyme (FLICE) [12] are involved in TNF-α-induced beta cell apoptosis.

MicroRNAs (miRNAs) are a group of short, non-coding RNAs (18 to 25 nucleotides) that may bind to complementary target sites in the 3′-untranslated regions (UTR) of mRNAs. These can inhibit gene expression post-transcriptionally, or can induce protein degradation in both animals and humans [13,14]. Many studies have identified miRNAs (miRs) as key regulators of beta cell proliferation, differentiation, insulin secretion, and apoptosis [15,16,17,18]. In IL-1β-treated beta cells, miR-101a and miR-30b were increased. These miRNAs contribute to beta cell dysfunction by decreasing insulin content, gene expression, and increasing cell death by targeting the Neurogenic differentiation 1 (NeuroD1) [16,19,20,21]. In MIN-6 (mouse beta cell line) cells, miR-34a is also involved in apoptosis pathways by directly binding to the 3′-UTR of the anti-apoptotic gene, *BCL2* (B-cell lymphoma 2) [22]. Therefore, understanding the role of miRNAs in IL-6-induced beta cell death will be beneficial in the pursuit of regulatory mechanisms involved in IL-6-induced apoptosis.

To identify mRNAs and miRNAs associated with IL-6-induced beta cell apoptosis, we investigated changes in gene expression in IL-6-treated INS-1 cells. In this study, we found that TNF-α expression was highly upregulated in IL-6-treated INS-1 cells, and that miR-181c contributed to IL-6-induced beta cell apoptosis through regulation of TNF-α expression.

## 2. Results

### 2.1. Induction of Apoptosis in IL-6 Treated Cells

To investigate induction of apoptosis by chronic IL-6 treatment, cell viability and annexin V- stained INS-1 cells were measured after 48 h treatment. As shown in Figure 1, we confirmed that treatment with 20 ng/mL of IL-6 increased early apoptotic cell populations, and cell viability was significantly reduced (Figure 1A,B).

### 2.2. Differential Gene Expression during Apoptosis in IL-6-Treated Cells

To identify apoptotic mechanisms activated in response to IL-6 treatment, differences in mRNA levels of apoptosis-related genes were examined using a custom RT^2^ profiler PCR array by comparing IL-6-treated cells with control, untreated cells. We observed significant upregulation or downregulation of many genes (Appendix A). A total of 26 genes were upregulated (>2-fold difference in expression) in IL-6 treated cells compared to untreated cells. Among them, expression levels of tumor necrosis factor (*TNF*)-α, toll-like receptor 2 (*Tlr2*), lympotoxin beta (*Ltb*), baculoviral IAP repeat-containing 3 (*Birc3*), Suppressor of cytokine signaling 3 (*Socs3*), Caspase 4, apoptosis-related cysteine peptidase (*Casp4*), insulin-like growth factor 1 receptor (*IGF1R*), and plasminogen (*Plg*) were the highest (>four-fold difference in expression) compared with control (Table 1). These genes were either proapoptotic (*TNF-α, Plg*, and *Casp4*), anti-apoptotic (*IGF1R* and *Birc3*) or involved in IL-6-mediated receptor signaling (*Tlr2* and *Socs3*).

### 2.3. Involvement of TNF-α Expression Regulation in IL-6-induced Beta Cell Apoptosis

As TNF-α levels showed the greatest increase (90-fold) during PCR array analysis, we tested the mRNA expression of TNF-α using qRT-PCR. We found that TNF-α mRNA levels were gradually increased by IL-6 treatment after 6, 12, and 24 h (Figure 2A). To investigate whether newly synthesized total proteins including TNF-α affected IL-6-induced beta cell apoptosis, cells were treated with 100 nM of cyclohexamide, a protein synthesis inhibitor, with or without 20 ng/mL of IL-6. Apoptotic populations were measured by annexin V-PE staining. As shown in Figure 2B, the number of annexin V-stained cells increased among IL-6-treated cells, and cyclohexamide treatment significantly reduced staining (Figure 2B). Finally, to confirm whether TNF-α was one of the proteins involved in IL-6-induced apoptosis, cells were pretreated with an anti-apoptotic, TNF-α neutralizing antibody at 50 μg/mL concentration in accordance with the manufacturer’s protocol. We observed that treatment with 50 μg/mL of TNF-α neutralizing antibody (NAb) reduced the population of annexin V-stained cells and, therefore, the apoptotic cells induced by chronic IL-6 treatment, compared with control Ab-treated cells (Figure 2C,D).

### 2.4. Downregulation of miR-181c during IL-6-Induced Beta Cell Apoptosis

To determine the involvement of miRNAs in IL-6-induced apoptosis, we analyzed global miRNA expression in INS-1 cells using Rat miRNome RT^2^ miRNA PCR array and miRDB (www.mirdb.org) prediction algorithm. We found that miR-101a, -122, and -181c were significantly downregulated about two-fold in IL-6-treated cells compared with control cells. To evaluate these results, we quantified miRNA expression using qRT-PCR. Among the three miRNAs, only the level of miR-181c was significantly decreased in INS-1 cells exposed to 20 ng/mL of IL-6 compared with untreated cells (Figure 3A). As our previous study showed that STAT-3 signaling mediated IL-6-induced beta-cell apoptosis, a STAT-3 inhibitor, AG490 [10], was used to determine whether miR-181c expression was regulated by STAT-3 inactivation. A treatment of 10 μM of AG490 effectively reduced STAT-3 phosphorylation in INS-1 cells [10] and downregulation of miR-181c by IL-6 treatment was reversed by co-treatment with AG490 (Figure 3B).

### 2.5. Inhibition of IL-6 Induced Beta Cell Cytotoxicity via miR-181c Upregulation

As it was reported that miR-181c is a new regulator of TNF-α expression [23], we tested whether miR-181c has an effect on apoptosis induced by IL-6. First, to investigate whether TNF-α-mediated beta cell apoptosis, in response to IL-6 treatment, is regulated by miR-181c expression, we examined the population of apoptotic cells in cells overexpressing miR-181c. We observed that cells transfected with a miR-181c mimic, ranging from ~1 to 20 pmol, showed elevated miR-181c levels. Cells transfected with 10 pmol or more showed a prominent increase in miR-181c levels (Figure 4A). Both mRNA and protein level of TNF-α were significantly decreased in the cells transfected with miR-181c mimic compared with those transfected with control miRNA (Figure 4B,C). The wild type (pMir-TNF-α-3′-UTR) or mutated (pMir-TNF-α-3′-UTR-muta) plasmid was cotransfected with the miR-181c mimic and luciferase activity was measured. After 48 h transfection, the luciferase activity of the miR-181c mimic group was significantly lower than that of the miR-control cells and the reduction was rescued in the mutated plasmid transfected cells (Figure 4D). Moreover, increased cytotoxicity by IL-6 treatment was significantly attenuated in cells overexpressing miR-181c (Figure 4E).

## 3. Discussion

Prolonged activation of inflammation is an important component of type 1 and type 2 diabetes progression, and its association with insulin secretion and apoptosis in beta cells has been well established [24,25,26]. One of the important mechanisms of beta cell damage during type 1 diabetes is increased expression of proinflammatory cytokines such as IL-1β, IFN-γ and TNF-α and cytokine induced beta cells were commonly used as in vitro model of type 1 diabetes [24,27]. Therefore, specific inhibitors targeting proinflammatory cytokines is a therapeutic strategy for treatment of type 1 diabetes [28].

IL-6 is one of the most abundant cytokines associated with metabolic disorders. Inhibition of IL-6 signaling, such as with an IL-6R antagonist tocilizumab, has become a valuable target during treatment of inflammatory disorders [29,30,31]. Previously, we found that chronic treatment of beta cells with IL-6 induced toxic effects through receptor-mediated STAT-3 signaling [10], but there is a need to further understand the molecular mechanisms of IL-6-induced apoptosis.

Among the key regulators involved in apoptosis, microarray analysis suggested that TNF-α is relevant to IL-6-induced apoptosis. It was reported that TNF-α expression was elevated in the blood of type 2 diabetic patients, resulting in an inflammatory response and insulin resistance [32]. During the development of type 2 diabetes, activated macrophages in the adipose tissues increased secretion of TNF-α expression, which induced the production of several inflammatory cytokines, hormones and their receptors and controls the mechanisms that induce metabolic stress [2]. Additionally, treatment of beta cells and pancreatic islet cells with TNF-α induces apoptosis, involving receptor-mediated caspase activation, NF-κB activation, and ceramide production [12,33]. Therefore, the addition of a neutralizing antibody against TNF-α with IL-6 treatment results in reduced annexin V staining in INS-1 cells.

Previously, we found that STAT-3-mediated nitric oxide (NO) production was involved in IL-6-induced beta cell apoptosis [10]. Chappell et al. identified the binding site for STAT-3 within the promoter region of TNF-α. Therefore, mutation of STAT-binding sites affected TNF-α expression [34]. These results suggested that the IL-6/STAT-3 pathway is responsible for the induction of TNF-α expression, and is consequently an important mediator of IL-6-induced apoptosis in INS-1 cells.

Recently, many miRNAs that are involved in beta cell apoptosis and gene expression were protective against cytokine-mediated cell death. In pancreatic islet cells from type 2 diabetic patients, miR-124a was highly expressed, and its silencing in MIN-6 cells resulted in increased expression of genes involved in insulin secretion, such as Mtpn, Foxa2, Sirt1, and NeuroD1 [35]. Moreover, exposure of MIN-6 cells or human islet cells to IL-1β and TNF-α increased miR-34a and miR-146a expression and reduction of these miRNAs was protective against cytokine-induced beta cell death [36]. miR-181c, a miRNA that is abundantly expressed in many tissues, has been implicated in a variety of diseases, including type 2 diabetes mellitus. Expression of miR-181c was decreased in the endothelial cells of type 2 diabetic db/db (leptin receptor mutation) mice, and in endothelial cell lines treated with high glucose levels compared with control [37]. Another study reported that serum miR-181c levels could be used as a biomarker for sensitive detection of diabetic retinopathy [38] and miR-181-3p and -5p were significantly downregulated in blood samples from diabetic patients compared with healthy patients [39]. Treatment with IL-6 decreased miR-181c expression, which is rescued by the addition of AG490, suggesting that decreased expression of miR-181c is a marker of IL-6-induced beta cell apoptosis.

MiR-181c and STAT-3 expression levels were inversely correlated in liver biopsy specimens [40]. In MCF10A breast cancer cells, putative STAT-3 binding sites were identified in the promoter of miR-181b-1 [41]. Although we did not investigate the direct interaction of STAT-3 and miR-181c in this study, these reports including our results suggested that the effect regarding IL6 mediated miR-181c downregulation was mediated by STAT-3 signaling and direct interaction of STAT-3 in the promoter region of miR-181c will be needed in future studies.

Although several targets of miR-181c have been identified including protein kinase C (PKC) delta, transforming growth factor beta (TGFB)-I, NAD-dependent deacetylase sirtuin (SIRT)-1, syntaxin 3, and homeobox protein Hox-A11 [42,43,44], targets related to beta cell apoptosis have not yet been reported. In this study, we demonstrated that miR-181c regulates TNF-α gene expression in association with beta cell apoptosis. It was reported that miR-181c directly regulates TNF-α expression by binding to its 3′-UTR in microglia cells [45], and a further study also demonstrated that TNF-α was the target of human miR-181c in hematopoietic progenitors [46]. These results in combination with our results, where miR-181c mimics decreased TNF-α expression and consequently the population of apoptotic cells, suggest that miR-181c regulation of TNF-α expression is a major mechanism involved in IL-6-induced apoptosis in beta cells.

Beside TNF-α/miR-181c, several (up- and downregulated-) genes and miRNAs were observed in IL-6 treated cells, the involvement of additional apoptotic mechanisms will be required. In conclusion, our data suggests that miR-181c regulation of TNF-α expression is an important mechanism involved in apoptotic pathways upon IL-6 treatment, and that upregulation of miR-181c expression may lead to inhibition of IL-6-induced apoptosis in beta cells (Figure 5). These findings also suggested that TNF-α inhibitors or upregulations of miR-181c expression may be a potential therapeutic target to preserve beta cell mass, at least in part, for the treatment of type 1 and type 2 diabetes and much further studies will be needed to apply clinical application.

## 4. Materials and Methods

### 4.1. Materials

The recombinant mouse IL-6 and TNF-α blocking antibody (MAB 510) was prepared by R&D systems (Minneapolis, MN, USA). RT^2^ profiler^TM^ PCR arrays were purchased from SABioscience Corp. (Germantown, MD, USA). The following reagents were purchased from the indicated suppliers: Annexin V-PE apoptosis detection kit (BD Transduction Laboratories, Palo Alto, CA, USA); RT^2^ first strand kit, RT^2^ qPCR master-mix, and miRNA-mimics (Qiagen Ltd., Germantown, MD, USA). All other chemical reagents were from Sigma Aldrich (St. Louis, MO, USA), Invitrogen (Carlsbad, CA, USA), or Takara (Takara, Shiga, Japan).

### 4.2. Cells

The rat insulinoma cell line, INS-1 (passage ~20 to 30), was grown in Roswell Park Memorial Institute (RPMI)-1640 medium containing 11.1 mmol/L glucose. The culture medium was supplemented with 10% fetal bovine serum (FBS), 2 mmol/L: Glutamine, 100 units/mL penicillin and 100 μg/mL streptomycin. For experiments, INS-1 cells were incubated with 20 ng/mL IL-6 for indicated time points.

### 4.3. Cell Viability Assay

Cells were treated with RPMI media containing 3-(4,5-dimethylthiazol-2-yl)-2,5-diphenyltetrazolium bromide (MTT; 0.5 mg/mL) for 4 h at 37 °C. Supernatants were discarded and isopropanol was then added. Cell viability was measured by absorbance at 570 nm using a microplate reader (Molecular Devices Corp., Menlo Park, CA, USA).

### 4.4. RNA Isolation and RT^2^ Profiler PCR Array

Total RNA extraction and purification were performed using the RNeasy MinElute kit (Qiagen Ltd.), according to the manufacturer’s instructions. DNase enzyme digestion was performed to exclude genomic DNA contamination. RNA was quantified using NanoDrop 1000 spectrophotometer (Thermo Scientific, Waltham, MA, USA) and the integrity of the sample was assessed in a 2100 Bioanalyzer (Agilent, Santa Clera, CA, USA). Each RNA sample was converted into cDNA using the RT^2^ first strand kit (Qiagen Ltd.), and cDNA sample was mixed with RT^2^ SYBR Green ROX^TM^ qPCR Mastermix (Qiagen Ltd.). Real-time PCR quantification was performed on an ABI 7900 Real-time PCR system (95 °C for 10 min, 40 cycles of 95 °C for 15 s and 60 °C for 1 min, followed by 1 cycle of 95 °C for 1 min, 55 °C for 30 s and 95 °C for 30 s) using an RT^2^ profiler custom PCR array consisting of a panel of 41 genes involved in apoptotic signaling, including a house-keeping gene (β-actin). Data were analyzed using the integrated web-based automated software for RT^2^ profiler PCR Array Data analysis (RT^2^ profiler PCR array Data analysis version 3.5, GeneGlobe Data Analysis, Qiagen Ltd.) available through SABiosciences. Fold changes in gene expression were calculated using the ^ΔΔ^C_t_ method, and β-actin was used for normalization of the results.

### 4.5. Quantitative Real-Time PCR

Extracted RNA was treated with DNase I and reverse-transcribed to single-strand cDNA using oligo(dT) primer with PrimeScript^TM^ RTase (Takara). Quantitative real-time PCR (qRT-PCR) analysis was performed using SYBR master mix (Applied Biosystems, Foster City, CA, USA) using the ABI 7900 Real-time PCR system according to the protocols provided by the manufacturer (Applied Biosystems). The sequences of the primer pairs are as follows: TNF-α (forward) 5′-CAG CCG ATT TGC CAT TTC A-3′ and (reverse) 5′-AGG GCT CTT GAT GGC AGA GA-3′; and cyclophilin (forward) 5′-GGTCTTTGGGAAGGTGAAAGAA-3′, and (reverse) 5′-GGTCTTTGGGAAGGTGAA AGAA-3′. The relative mRNA transcript levels were calculated according to the 2Δ^–CT^ method, in which Δ*C*T represents the difference in threshold cycle values between the target mRNA and the cyclophilin internal control.

### 4.6. miRNA Isolation and Quantitative RT-PCR

Total RNA from INS-1 cells treated with 20 ng/mL IL-6 was extracted using the Qiagen miRNA isolation kit. Extracted miRNA was reverse transcribed to single-strand cDNA with RT^2^ miRNA first strand kit (Qiagen). Real-time quantitative PCR analysis (Rat miRNome RT^2^ miRNA PCR array, Qiagen) was performed using SYBR master mix using the ABI 7900 Real-time PCR system according to the protocols provided by the manufacturer (Applied Biosystems). Primer sequence was as follows: miR-101a (rno miR-101a miScript Primer assay, Qiagen cat# MS00012950), miR-122a (rno miR-122a miScript Primer assay, Qiagen cat# MS00000315), and miR-181c (rno miR-181c miScript Primer assay, Qiagen cat# MS MS00013132). The relative mRNA transcript levels were calculated according to 2Δ^–CT^ method, in which ΔCT represents the difference in threshold cycle values between the target miRNA and Rnu6, which was used as an internal control.

### 4.7. Annexin V-PE Staining

Early apoptotic cells were determined using an Annexin V-PE apoptosis detection kit, according to the manufacturer’s instructions. Briefly, following IL-6 treatment, cells were harvested and washed twice with PBS. The cells were then suspended in 1× binding buffer, and annexin V-PE was added. After incubation for 15 min, stained cells were analyzed by flow cytometry (FACS Calibur) using CELLQuest Pro software (Version 5.1, BD Biosciences, Franklin Lakes, NJ, USA).

### 4.8. Transfection of MicroRNA Mimics

INS-1 cells were transfected with miR-181c (miR-181c-5p; Qiagen cat#MSY0000857) mimics or miR-control using Lipofectamine 2000 (Invitrogen) as per the manufacturer’s protocol. After 24 h transfection, the medium was replaced with fresh medium containing 20 ng/mL IL-6.

### 4.9. Luciferase Reporter Assay

HEK293 cells were transfected with pMir-TNF-α-3′-UTR (OriGene Technologies, Inc. Rockville, MD, USA), pMir-TNF-α-3′-UTR mutant (OriGene, mutate nucleotides 532–538 TGAATGT to ACTTACA), and miR-181c mimic using Lipofectamine 2000. Cells were collected 48 h after transfection and luciferase activity was measured using a luciferase reporter assay system (Promega, Madison, WI, USA).

### 4.10. Statistical Analysis

Data are presented as mean ± standard error of mean (SEM) of three independent experiments. Data were analyzed using analysis of variance (ANOVA) followed by post hoc analysis using the Tukey range test (SPSS 10.0 statistical software, SPSS Inc., Chicago, IL, USA). An unpaired 2-tailed *t*-test was used for analysis of the 2 groups. *p* < 0.05 was considered to indicate a statistically significant difference between values.

## Figures and Tables

**Figure 1 molecules-24-01410-f001:**
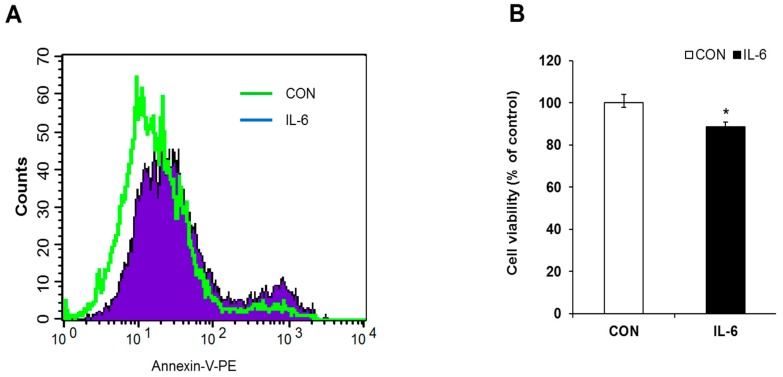
Effect of Interleukin (IL)-6 on apoptosis and cell viability in beta cells. INS-1 cells were treated with 20 ng/mL IL-6 for 48 h and (**A**) stained with FITC-annexin V/PI and analyzed by flow cytometry to determine the population of cells in early apoptosis. (**B**) Cells were treated as described in (**A**), and cell viability was determined by MTT assay. The results represent the mean ± SEM from experiments performed in triplicate and normalized to control (CON) cells. * *p* < 0.05 in comparison with CON.

**Figure 2 molecules-24-01410-f002:**
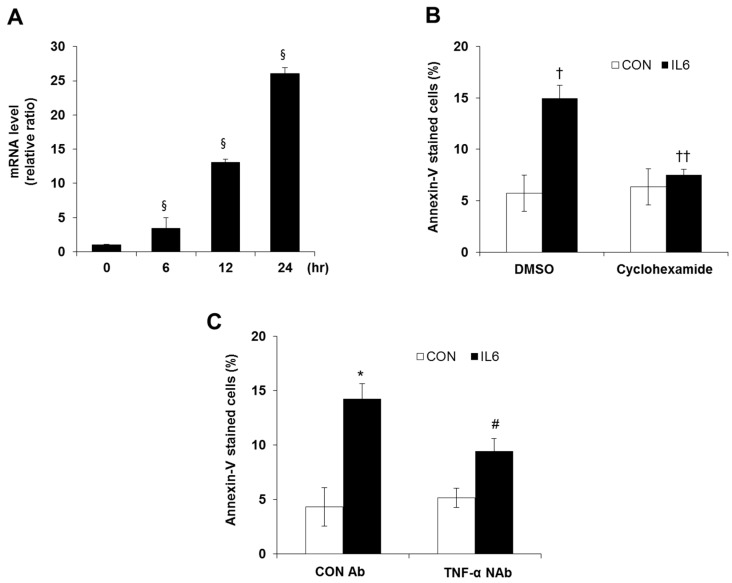
TNF-α mRNA expression changes in IL-6-induced beta cell apoptosis. (**A**) INS-1 cells were treated with 20 ng/mL of IL-6 for indicated time points and TNF-α mRNA levels were analyzed by quantitative RT-PCR. The mRNA levels were normalized with those of cyclophilin. (**B**) The cells were preincubated with cyclohexamide (3 h, 100 nM) and then incubated in the presence or absence of 20 ng/mL IL-6. Annexin-V-stained cells were analyzed by flow cytometry. (**C**) The cells were preincubated with 50 μg/mL control antibody (mouse IgG Ab) or neutralizing antibody (NAb) against TNF-α for 1 h and then incubated in the presence or absence of 20 ng/mL IL-6. Annexin-V stained cells were analyzed by flow cytometry. The results shown represent the mean ± SEM from experiments performed in triplicate. ^§^
*p* < 0.05 in comparison with 0 h, † *p* < 0.05 in comparison with DMSO treated CON, †† *p* < 0.05 in comparison with DMSO treated with IL6, * *p* < 0.05 in comparison with CON Ab treated CON, # *p* < 0.05 in comparison with CON Ab treated with IL6.

**Figure 3 molecules-24-01410-f003:**
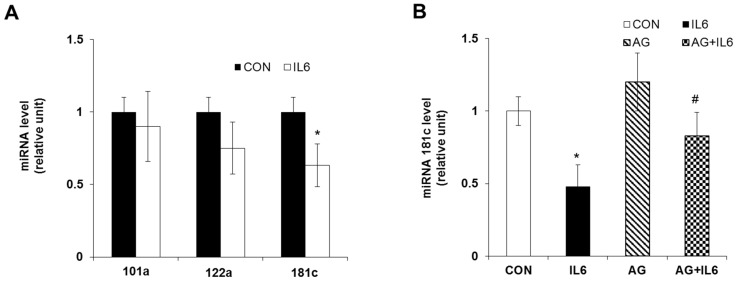
Downregulation of miR-181c during IL-6-induced beta cell apoptosis. (**A**) INS-1 cells were treated with 20 ng/mL IL-6 for 24 h and miRNA levels were analyzed by quantitative RT-PCR. (**B**) INS-1 cells were pretreated with or without AG490 (10 μM) for 3 h and then incubated with 20 ng/mL IL-6 for 24 h. miR-181c levels were analyzed by quantitative RT-PCR and normalized to endogenous RNU6. The results shown represent the mean ± SEM from experiments performed in triplicate. * *p* < 0.05 in comparison with CON, ^#^
*p* < 0.05 in comparison with IL6.

**Figure 4 molecules-24-01410-f004:**
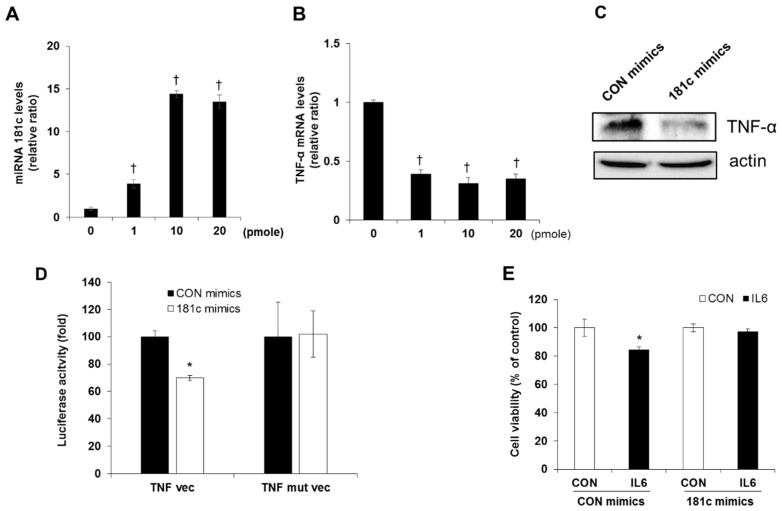
Inhibition of IL-6-induced beta cell apoptosis via miR-181c upregulation. INS-1 cells were treated with miR-181c mimics for 24 h. Levels of (**A**) miR-181c, (**B**) TNF-α mRNA, and (**C**) TNF-α protein level were analyzed by quantitative RT-PCR and Western blot analysis. (**D**) HEK293 cells were transfected with each of the constructed plasmids, together with control mimic and miR-181c mimic and luciferase activity was measured. (**E**) INS-1 cells were treated with 10 pmol of miR-181c mimics for 24 h and then incubated in the presence or absence of 20 ng/mL IL-6 for 48 h. Cell viability was determined by MTT assay. The results represent the mean ± SEM from experiments performed in triplicate and normalized to control (CON) cells. ^†^
*p* < 0.05 in comparison with 0 pmole, * *p* < 0.05 in comparison with CON mimics treated CON.

**Figure 5 molecules-24-01410-f005:**
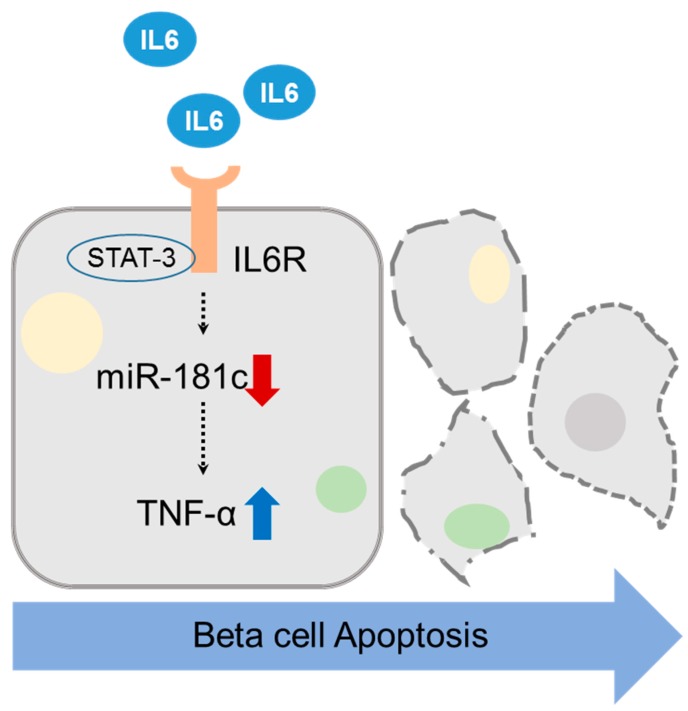
Schematic diagram of the pathway involved in IL-6-induced beta cell apoptosis mediated by miR-181c/TNF-α in INS-1 cells.

**Table 1 molecules-24-01410-t001:** Fold changes of apoptotic gene expression in IL-6-treated beta cells compared to untreated cells.

Gene Symbol	Name of Gene	Fold Up-Regulation (IL-6-Treated vs. Control)	*p*-Value
*TNF*	Tumor necrosis factor (TNF superfamily, member 2)	92.7929	0.001
*Tlr2*	Toll-like receptor 2	21.3735	0.009
*Ltb*	Lymphotoxin beta (TNF superfamily, member 3)	17.7839	0.003
*Birc3*	Baculoviral IAP repeat-containing 3	11.4395	0.016
*Socs3*	Suppressor of cytokine signaling 3	9.4656	0.01
*Casp4*	Caspase 4, apoptosis-related cysteine peptidase	7.9376	0.009
*Igf1r*	Insulin-like growth factor 1 receptor	5.9825	0.004
*Plg*	Plasminogen	4.1543	0.001

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
