# Peer review of "MicroRNA-181c Inhibits Interleukin-6-mediated Beta Cell Apoptosis by Targeting TNF-α Expression"

_molecules, 2019, doi:10.3390/molecules24071410_

Round 1

Reviewer 1 Report

The manuscript by Oh et al. describes the direct and post transcriptional inhibition of TNFalpha by miR-181c upon interleukin-6 treatment and demonstrates that this interaction reduces apoptotic response induced by IL6.

The manuscript is written in a fluent English and MM section provides an highly detailed description of experimental procedures exploited along the whole paper. Moreover, results are nicely presented and in my opinion well support the conclusions provided by the authors.

Nonetheless, in my opinion, some corrections and/or details extra might definitely improved quality of this manuscript.

although TNFalpha is the more intensely upregulated gene upon IL6 treatment, the complete list of deregulated genes might be included, even in a supplementary table.

I suggest to include some further details about miRNA microarray analysis cited at the beginning in paragraph 2.4. Is it an unpublished experiment exploited by the authors or an analysis run on literature data and/or GEO repository?

line 194 "promoter" instead of "promoters".

line 200 "dependent" instead of " depnedent"

in line 49 I would suggest a more fluent "...to complementary target sites in the 3'UTR of target mRNAs"

Author Response

[Molecules-475807] MicroRNA-181c inhibits interleukin-6-mediated beta cell apoptosis by targeting TNFα expression

Yoon Sin Oh, Gong Deuk Bae, Eun-Young Park and Hee-Sook Jun

The manuscript by Oh et al. describes the direct and post transcriptional inhibition of TNFalpha by miR-181c upon interleukin-6 treatment and demonstrates that this interaction reduces apoptotic response induced by IL6.

The manuscript is written in a fluent English and MM section provides an highly detailed description of experimental procedures exploited along the whole paper. Moreover, results are nicely presented and in my opinion well support the conclusions provided by the authors.

Nonetheless, in my opinion, some corrections and/or details extra might definitely improved quality of this manuscript.

1.     although TNFalpha is the more intensely upregulated gene upon IL6 treatment, the complete list of deregulated genes might be included, even in a supplementary table.

Response: Thanks for your valuable comment. As the reviewer’s comments, we included the list of deregulated genes in the supplementary table of revised manuscript (Supplementary Table 1, line 79).

2.     I suggest to include some further details about miRNA microarray analysis cited at the beginning in paragraph 2.4. Is it an unpublished experiment exploited by the authors or an analysis run on literature data and/or GEO repository?

Response: miRNA microarray was performed using Rat miRNome RT2 miRNA PCR array (Qiagen). The Rat miRNome RT2 miRNA PCR array profiles the expression of the 370 most abundantly expressed and best characterized miRNA sequences in the rat genome as annotated by the Sanger miRBase Release 14. Among the deregulated miRNAs, predicted miRNAs to regulate TNF-α mRNA was checked using miRDB (www.mirdb.org) prediction algorithm. 3 miRNAs were found as important regulators of TNF-α expression and we validated the changes of expression by qRT-PCR analysis in IL-6 treated cells. We added this information in the revised manuscript (line 118, 267).

3.     line 194 "promoter" instead of "promoters".

Response: We changed to ‘promoter’.

4.     line 200 "dependent" instead of " depnedent".

Response: We changed to ‘dependent’.

5.     in line 49 I would suggest a more fluent "...to complementary target sites in the 3'UTR of target mRNAs"

Response: As the reviewer’s suggestion, we changed the sentence in the revised manuscript (line 49).

Reviewer 2 Report

The authors have done a nice job in addressing my comments

Author Response

Thank you for review.

This manuscript is a resubmission of an earlier submission. The following is a list of the peer review reports and author responses from that submission.

Round 1

Reviewer 1 Report

This manuscript by Oh et al., examines the expression changes in beta cells exposed to IL6 stimulation over time. Overall I think that the data and discussion of this work is nicely presented and a pleasure to read. There are a few minor corrections that would really improve this manuscript. They are listed below:

Figures:

1) Statistical analysis of 2A

2) Statistical analysis of 3B comparing IL6 vs. AG+IL6

3) Statistical analysis of 4A and 4B

Text Changes:

Page 43: Full names TNFR1 and @

49: replace DNA with RNA

53/55: more references with both reference 10 and 11.

55: explanation of MIN-6 cells

55: full name: NeuroD

67: added "INS-1" after stained

81: change "several" to "many"

94: change "mostly increased" to " showed the greatest increase"

95: changed "validated" to "tested the"

123: change "validate" to evaluate

124: add "only" after miRNAs

158/160: more references with 13 and 14

183: change "an" to "a"

185/187/188/200: please do not lead sentences with authors names. This is not a good scientific standard.

186: explain db/db mice

194: change "validated such as" to identified including"

196: change "but the" to "no"

205: "various" is not a scientific word, please change.

206: add "additional" after of

237: English is poor, correct tense

239: replace RNase with DNase

With these minor corrections, I think this manuscript will be much easier to read and of value to the scientific community.

Reviewer 2 Report

The manuscript by Oh YS et al. aims to demonstrate that miR-181c interferes with the IL6-induced apoptosis of beta cells by targeting TNFalpha signaling.

TNFalpha regulation by miR-181 family members had already been reported in other biological context by Zhu et al. in 2017. At the same time, mir-181c down-regulation by IL-6 has been very recently described by Shen et al. Together these previous works confer only a moderate novelty to this work.

The whole manuscript is written in a fluent English and the experimental procedures are highly detailed.The results are also well presented but some interpretations should be corroborated by additional analysis and controls.

Specific comments as follows (some very general, numbered for simplicity):

Minor comments:

#1- Duration of IL-6 treatment is differently indicated: 48 hrs in line 67, 24hrs in line 72 and in the rest of the manuscript. How long is the duration of IL6 treatment after AG490 pretreatment (line 134) and after miR-181c transfection?

#2-  line 133 – concentration of the antibodies could be indicated together at the beginning if both the control and the neutralizing antibodies are used in the same manner.

#3 - line 117 - #P describes statistics of Fig. 2C whereas #P of Fig 2B is missing.

#4 - line 124 – I would add “ONLY the level of miR-181c was significantly……” underlying that solely the downregulation of miR-181c could be confirmed by qRT-PCR.

#5-  line 150 –cite panel 4C in the legend.

#6 - line 196 - remove "but the"

#7 - line 198 - regulates instead of "regulated".

#8 - line 239 – Dnase instead of “RNAse” digestion.

Major regards:

#1 – Authors should provide explanation about why they exploited INS-1 insulinoma cells as the model of choice. Do these cells respond to IL6 treatment as healthy cells do?

#2 – Are there any other mRNA affected by miR-181c among the the list found deregulated upon IL6 treatment,?

#3 – Figure 2 – CHX experiment provides, in my opinion, only information about the role of newly-synthesized total proteins during IL6-induced apoptosis, without any indication on the specific TNFalpha contribution as instead reported by the authors in line 97. Is this in order to underlie the autocrine effect of the newly synthesized TNF?

In addition, are there any available controls of the occurred TNFalpha neutralization?

#4 - The authors demonstrated the role of IL-6 treatment on miR-181c downregulation by mean of an inhibitor of the STAT3 signalling. But is it effect direct or mediated in turn by other mediators? Is it a repressive effect via any STAT3 putative binding sites in the promoter region of miR-181c? 

Are there any positive control of the effective STAT3 inhibition?

#5 - Analogously, do miR-181c directly target TNFalpha mRNA? Zhu et al already described this interaction but I strongly suggest luciferase reporter assay with the wildtype and mutated forms of the 3’UTR of TNFalpha mRNA in order to demonstrate the direct nature of this regulation. 

#6 - I suggest to detect TNFalpha at the protein level by immunoblotting /WB after miR-181c transfection (Fig.4).

#7- line 144 - Fig 4C, obtained by MTT assay, provides information only about cell viability so any speculation on apoptosis should be removed.

#8 - did the authors transfect miR-181c-3p or -5p?

#9 - the diagram shown in Fig 5 depicts very clearly the proposed molecular mechanism. Nonetheless, if I understood correctly, apoptosis is triggered in an autocrine manner so the term 'apopstosis' should be applied to neighbouring cells.